# High-Dimensional Multivariate Forecasting with Low-Rank Gaussian Copula Processes

**David Salinas**
Naverlabs *
david.salinas@naverlabs.com

**Michael Bohlke-Schneider**
Amazon Research
bohlkem@amazon.com

**Laurent Callot**
Amazon Research
lcallot@amazon.com

**Roberto Medico**
Ghent University *
roberto.medico91@gmail.com

**Jan Gasthaus**
Amazon Research
gasthaus@amazon.com

## Abstract

Predicting the dependencies between observations from multiple time series is critical for applications such as anomaly detection, financial risk management, causal analysis, or demand forecasting. However, the computational and numerical difficulties of estimating time-varying and high-dimensional covariance matrices often limits existing methods to handling at most a few hundred dimensions or requires making strong assumptions on the dependence between series. We propose to combine an RNN-based time series model with a Gaussian copula process output model with a low-rank covariance structure to reduce the computational complexity and handle non-Gaussian marginal distributions. This permits to drastically reduce the number of parameters and consequently allows the modeling of time-varying correlations of thousands of time series. We show on several real-world datasets that our method provides significant accuracy improvements over state-of-the-art baselines and perform an ablation study analyzing the contributions of the different components of our model.

## 1 Introduction

The goal of forecasting is to predict the distribution of future time series values. Forecasting tasks frequently require predicting several related time series, such as multiple metrics for a compute fleet or multiple products of the same category in demand forecasting. While these time series are often dependent, they are commonly assumed to be (conditionally) independent in high-dimensional settings because of the hurdle of estimating large covariance matrices.

Assuming independence, however, makes such methods unsuited for applications in which the correlations between time series play an important role. This is the case in finance, where risk minimizing portfolios cannot be constructed without a forecast of the covariance of assets. In retail, a method providing a probabilistic forecast for different sellers should take competition relationships and cannibalization effects into account. In anomaly detection, observing several nodes deviating from their expected behavior can be cause for alarm even if no single node exhibits clear signs of anomalous behavior.

Multivariate forecasting has been an important topic in the statistics and econometrics literature. Several multivariate extensions of classical univariate methods are widely used, such as vector autoregressions (VAR) extending autoregressive models [19], multivariate state-space models [7], or

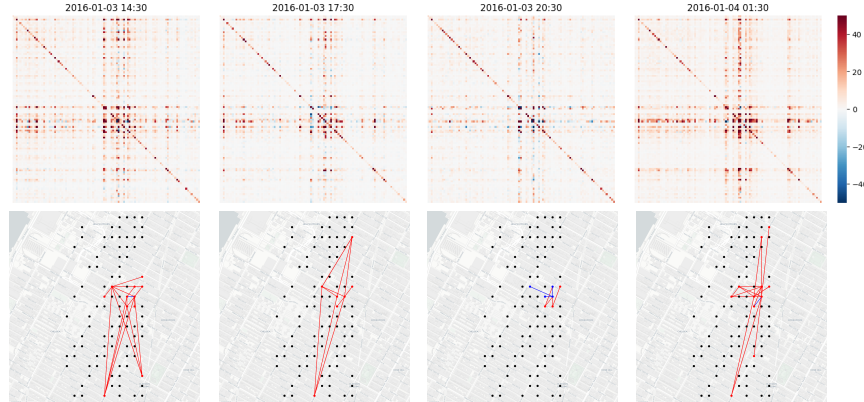

Figure 1: Top: covariance matrix predicted by our model for taxi traffic time series for 1214 locations in New-York at 4 different hours of a Sunday (only a neighborhood of 103 series is shown here, for clearer visualization). Bottom: Correlation graph obtained by keeping only pairs with covariance above a fixed threshold at the same hours. Both spatial and temporal relations are learned from the data as the covariance evolves over time and edges connect locations that are close to each other.

multivariate generalized autoregressive conditional heteroskedasticity (MGARCH) models [2]. The rapid increase in the difficulty of estimating these models due to the growth in number of parameters with the dimension of the problem have been binding constraints to move beyond low-dimensional cases. To alleviate these limitations, researchers have used dimensionality reduction methods and regularization, see for instance [3, 5] for VAR models and [34, 9] for MGARCH models, but these models remain unsuited for applications with more than a few hundreds dimensions [23].

Forecasting can be seen as an instance of sequence modeling, a topic which has been intensively studied by the machine learning community. Deep learning-based sequence models have been successfully applied to audio signals [33], language modeling [13, 30], and general density estimation of univariate sequences [22, 21]. Similar sequence modeling techniques have also been used in the context of forecasting to make probabilistic predictions for collections of real or integer-valued time series [26, 36, 16]. These approaches fit a global (i.e. shared) sequence-to-sequence model to a collection of time series, but generate statistically independent predictions. Outside the forecasting domain, similar methods have also been applied to (low-dimensional) multivariate dependent time series, e.g. two-dimensional time series of drawing trajectories [13, 14].

Two main issues prevent the estimation of high-dimensional multivariate time series models. The first one is the $O(N^2)$ scaling of the number of parameters required to express the covariance matrix where $N$ denotes the dimension. Using dimensionality reduction techniques like PCA as a pre-processing step is a common approach to alleviate this problem, but it separates the estimation of the model from the preprocessing step, leading to decreased performance. This motivated [27] to perform such a factorization jointly with the model estimation. In this paper we show how the low rank plus diagonal covariance structure of the *factor analysis* model [29, 25, 10, 24] can be used in combination with Gaussian copula processes [37] to an LSTM-RNN [15] to jointly learn the temporal dynamics and the (time-varying) covariance structure, while significantly reducing the number of parameters that need to be estimated.

The second issue affects not only multivariate models, but all global time series models, i.e. models that estimate a single model for a collection of time series: In real-world data, the magnitudes of the time series can vary drastically between different series of the same data set, often spanning several orders of magnitude. In online retail demand forecasting, for example, item sales follow a power-law distribution, with a large number of items selling only a few units throughout the year, and a few popular items selling thousands of units per day [26]. The challenge posed by this for estimating global models across time series has been noted in previous work [26, 37, 27]. Several approaches have been proposed to alleviate this problem, including simple, fixed invertible transformations such as the square-root or logarithmic transformations, and the data-adaptive Box-Cox transform [4],

that aims to map a potentially heavy-tailed distribution to a Normal distribution. Other approaches includes removing scale with mean-scaling [26], or with a separate network [27].

Here, we propose to address this problem by modeling each time series' marginal distribution separately using a non-parametric estimate of its cumulative distribution function (CDF). Using this CDF estimate as the marginal transformation in a Gaussian copula (following [18, 17, 1]) effectively addresses the challenges posed by scaling, as it decouples the estimation of marginal distributions from the temporal dynamics and the dependency structure.

The work most closely related to ours is the recent work [32], which also proposes to combine deep autoregressive models with copula to model correlated time series. Their approach uses a nonparametric estimate of the copula, whereas we employ a Gaussian copula with low-rank structure that is learned jointly with the rest of the model. The nonparametric copula estimate requires splitting a $N$-dimensional cube into $\varepsilon^{-N}$ many pieces (where $N$ is the time series dimension and $\varepsilon$ is a desired precision), making it difficult to scale that approach to large dimensions. The method also requires the marginal distributions and the dependency structure to be time-invariant, an assumption which is often violated is practice as shown in Fig. 1. A concurrent approach was proposed in [35] which also uses Copula and estimates marginal quantile functions with the approach proposed in [11] and models the Cholesky factor as the output of a neural network. Two important differences are that this approach requires to estimate $O(N^2)$ parameters to model the covariance matrix instead of $O(N)$ with the low-rank approach that we propose, another difference is the use of a non-parametric estimator for the marginal quantile functions.

The main contributions of this paper are:

- a method for probabilistic high-dimensional multivariate forecasting (scaling to dimensions up to an order of magnitude larger than previously reported in [23]),
- a parametrization of the output distribution based on a low-rank-plus-diagonal covariance matrix enabling this scaling by significantly reducing the number of parameters,
- a copula-based approach for handling different scales and modeling non-Gaussian data,
- an empirical study on artificial and six real-world datasets showing how this method improves accuracy over the state of the art while scaling to large dimensions.

The rest of the paper is structured as follows: In Section 2 we introduce the probabilistic forecasting problem and describe the overall structure of our model. We then describe how we can use the empirical marginal distributions in a Gaussian copula to address the scaling problem and handle non-Gaussian distributions in Section 3. In Section 4 we describe the parametrization of the covariance matrix with low-rank-plus-diagonal structure, and how the resulting model can be viewed as a low-rank Gaussian copula process. Finally, we report experiments with real-world datasets that demonstrate how these contributions combine to allow our model to generate correlated predictions that outperform state-of-the-art methods in terms of accuracy.

## 2 Autoregressive RNN Model for Probabilistic Multivariate Forecasting

Let us denote the values of a multivariate time series by $z_{i,t} \in \mathcal{D}$, where $i \in \{1, 2, \dots, N\}$ indexes the individual univariate component time series, and $t$ indexes time. The domain $\mathcal{D}$ is assumed to be either $\mathbb{R}$ or $\mathbb{N}$. We will denote the multivariate observation vector at time $t$ by $\mathbf{z}_t \in \mathcal{D}^N$. Given $T$ observations $\mathbf{z}_1, \dots, \mathbf{z}_T$, we are interested in forecasting the future values for $\tau$ time units, i.e. we want to estimate the joint conditional distribution $P(\mathbf{z}_{T+1}, ..., \mathbf{z}_{T+\tau} | \mathbf{z}_1, \dots, \mathbf{z}_T)$. In a nutshell, our model takes the form of a non-linear, deterministic state space model whose state $\mathbf{h}_{i,t} \in \mathbb{R}^k$ evolves independently for each time series $i$ according to transition dynamics $\varphi$,

$$\mathbf{h}_{i,t} = \varphi_{\theta_h}(\mathbf{h}_{i,t-1}, z_{i,t-1}) \qquad i = 1, \dots, N, \tag{1}$$

where the transition dynamics $\varphi$ are parametrized using a LSTM-RNN [15]. Note that the LSTM is unrolled for each time series separately, but parameters are tied across time series. Given the state values $\mathbf{h}_{i,t}$ for all time series $i = 1, 2, \dots, N$ and denoting by $\mathbf{h}_t$ the collection of state values for all series at time $t$, we parametrize the joint emission distribution using a Gaussian copula,

$$p(\mathbf{z}_t | \mathbf{h}_t) = \mathcal{N}([f_1(z_{1,t}), f_2(z_{2,t}), \dots, f_N(z_{N,t})]^T \mid \boldsymbol{\mu}(\mathbf{h}_t), \Sigma(\mathbf{h}_t)). \tag{2}$$

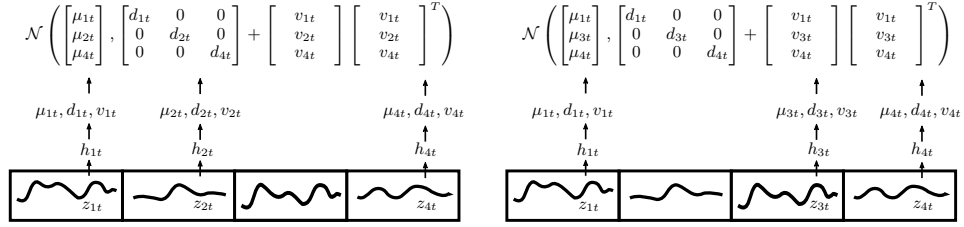

Figure 2: Illustration of our model parametrization. During training, dimensions are sampled at random and a local LSTM is unrolled on each of them individually (1, 2, 4, then 1, 3, 4 in the example). The parameters governing the state updates and parameter projections are shared for all time series. This parametrization can express the Low-rank Gaussian distribution on sets of series that varies during training or prediction.

The transformations $f_i : \mathcal{D} \to \mathbb{R}$ here are invertible mappings of the form $\Phi^{-1} \circ \hat{F}_i$, where $\Phi$ denotes the cumulative distribution function of the standard normal distribution, and $\hat{F}_i$ denotes an estimate of the marginal distribution of the $i$-th time series $z_{i,1}, \ldots, z_{i,T}$. The role of these functions $f_i$ is to transform the data for the $i$-th time series such that marginally it follows a standard normal distribution. The functions $\boldsymbol{\mu}(\cdot)$ and $\Sigma(\cdot)$ map the state $\mathbf{h}_t$ to the mean and covariance of a Gaussian distribution over the transformed observations (described in more detail in Section 4).

Under this model, we can factorize the joint distribution of the observations as

$$p(\mathbf{z}_1, \ldots \mathbf{z}_{T+\tau}) = \prod_{t=1}^{T+\tau} p(\mathbf{z}_t | \mathbf{z}_1, \ldots, \mathbf{z}_{t-1}) = \prod_{t=1}^{T+\tau} p(\mathbf{z}_t | \mathbf{h}_t). \tag{3}$$

Both the state update function $\varphi$ and the mappings $\boldsymbol{\mu}(\cdot)$ and $\Sigma(\cdot)$ have free parameters that are learned from the data. We denote $\theta$ the vector of all free parameters, consisting of the parameters of the state update $\theta_h$ as well as $\theta_{\boldsymbol{\mu}}$ and $\theta_{\Sigma}$ which denote the free parameters in $\boldsymbol{\mu}(\cdot)$ and $\Sigma(\cdot)$. Given $\theta$ and $\mathbf{h}_{T+1}$, we can produce Monte Carlo samples from the joint predictive distribution

$$p(\mathbf{z}_{T+1}, \ldots \mathbf{z}_{T+\tau} | \mathbf{z}_1, \ldots, \mathbf{z}_T) = p(\mathbf{z}_{T+1}, \ldots \mathbf{z}_{T+\tau} | \mathbf{h}_{T+1}) = \prod_{t=T+1}^{T+\tau} p(\mathbf{z}_t | \mathbf{h}_t) \tag{4}$$

by sequentially sampling from $P(\mathbf{z}_t | \mathbf{h}_t)$ and updating $\mathbf{h}_t$ using $\varphi$ [2]. We learn the parameters $\theta$ from the observed data $\mathbf{z}_1, \ldots, \mathbf{z}_T$ using maximum likelihood estimation by i.e. by minimizing the loss function

$$-\log p(\mathbf{z}_1, \mathbf{z}_2, \ldots, \mathbf{z}_T) = -\sum_{t=1}^{T} \log p(\mathbf{z}_t | \mathbf{h}_t), \tag{5}$$

using stochastic gradient descent-based optimization. To handle long time series, we employ a data augmentation strategy which randomly samples fixed-size slices of length $T' + \tau$ from the time series during training, where we fix the *context length* hyperparameter $T'$ to $\tau$. During prediction, only the last $T'$ time steps are used in computing the initial state for prediction.

## 3 Gaussian Copula

A copula function $C : [0,1]^N \to [0,1]$ is the CDF of a joint distribution of a collection of real random variables $U_1, \ldots, U_N$ with uniform marginal distribution [8], i.e.

$$C(u_1, \ldots, u_N) = P(U_1 \leq u_1, \ldots, U_N \leq u_N).$$

Sklar's theorem [28] states that any joint cumulative distribution $F$ admits a representation in terms of its univariate marginals $F_i$ and a copula function $C$,

$$F(z_1, \ldots, z_N) = C(F_1(z_1), \ldots, F_N(z_N)).$$

When the marginals are continuous the copula $C$ is unique and is given by the joint distribution of the *probability integral transforms* of the original variables, i.e. $\mathbf{u} \sim C$ where $u_i = F_i(z_i)$. Furthermore, if $z_i$ is continuous then $u_i \sim \mathcal{U}(0,1)$.

A common modeling choice for $C$ is to use the Gaussian copula, defined by:

$$C(F_1(z_1), \ldots, F_d(z_N)) = \phi_{\boldsymbol{\mu},\Sigma}(\Phi^{-1}(F_1(z_1)), \ldots, \Phi^{-1}(F_N(z_N))),$$

where $\Phi : \mathbb{R} \to [0,1]$ is the CDF of the standard normal and $\phi_{\boldsymbol{\mu},\Sigma}$ is a multivariate normal distribution parametrized with $\boldsymbol{\mu} \in \mathbb{R}^N$ and $\Sigma \in \mathbb{R}^{N \times N}$. In this model, the observations $\mathbf{z}$, the marginally uniform random variables $\mathbf{u}$ and the Gaussian random variables $\mathbf{x}$ are related as follows:

$$\mathbf{x} \xrightarrow{\Phi} \mathbf{u} \xrightarrow{F^{-1}} \mathbf{z} \qquad\qquad \mathbf{z} \xrightarrow{F} \mathbf{u} \xrightarrow{\Phi^{-1}} \mathbf{x}.$$

Setting $f_i = \Phi^{-1} \circ \hat{F}_i$ results in the model in Eq. (2).

The marginal distributions $F_i$ are not given a priori and need to be estimated from data. We use the non-parametric approach of [18] proposed in the context of estimating high-dimensional distributions with sparse covariance structure. In particular, they use the empirical CDF of the marginal distributions,

$$\hat{F}_i(v) = \frac{1}{m}\sum_{t=1}^{m} \mathbb{1}_{z_{it} \leq v},$$

where $m$ observations are considered. As we require the transformations $f_i$ to be differentiable, we use a linearly-interpolated version of the empirical CDF resulting in a piecewise-constant derivative $\hat{F}'(\mathbf{u})$. This allow us to write the log-density of the original observations under our model as

$$\begin{aligned}
\log p(\mathbf{z}; \boldsymbol{\mu}, \Sigma) &= \log \phi_{\boldsymbol{\mu},\Sigma}(\Phi^{-1}(\hat{F}(\mathbf{z}))) + \log \frac{d}{d\mathbf{z}}\Phi^{-1}(\hat{F}(\mathbf{z})) \\
&= \log \phi_{\boldsymbol{\mu},\Sigma}(\Phi^{-1}(\hat{F}(\mathbf{z}))) + \log \frac{d}{d\mathbf{u}}\Phi^{-1}(\mathbf{u}) + \log \frac{d}{d\mathbf{z}}\hat{F}(\mathbf{z}) \\
&= \log \phi_{\boldsymbol{\mu},\Sigma}(\Phi^{-1}(\hat{F}(\mathbf{z}))) - \log \phi(\Phi^{-1}(\hat{F}(\mathbf{z})) + \log \hat{F}'(\mathbf{z})
\end{aligned}$$

which are the individual terms in the total loss (5) where $\phi$ is the probability density function of the standard normal.

The number of past observations $m$ used to estimate the empirical CDFs is an hyperparameter and left constant in our experiments with $m = 100$ [3].

## 4 Low-rank Gaussian Process Parametrization

After applying the marginal transformations $f_i(\cdot)$ our model assumes a joint multivariate Gaussian distribution over the transformed data. In this section we describe how the parameters $\boldsymbol{\mu}(\mathbf{h}_t)$ and $\Sigma(\mathbf{h}_t)$ of this emission distribution are obtained from the LSTM state $\mathbf{h}_t$.

We begin by describing how a low-rank-plus-diagonal parametrization of the covariance matrix can be used to keep the computational complexity and the number of parameters manageable as the number of time series $N$ grows. We then show how, by viewing the emission distribution as a time-varying low-rank Gaussian Process $g_t \sim \mathrm{GP}(\tilde{\mu}_t(\cdot), k_t(\cdot, \cdot))$, we can train the model by only considering a subset of time series in each mini-batch further alleviating memory constraints and allowing the model to be applied to very high-dimensional sets of time series.

Let us denote the vector of transformed observations by

$$\mathbf{x}_t = f(\mathbf{z}_t) = [f_1(z_{1,t}), f_2(z_{2,t}), \ldots, f_N(z_{N,t})]^T,$$

so that $p(\mathbf{x}_t|\mathbf{h}_t) = \mathcal{N}(\mathbf{x}_t|\boldsymbol{\mu}(\mathbf{h}_t), \Sigma(\mathbf{h}_t))$. The covariance matrix $\Sigma(\mathbf{h}_t)$ is a $N \times N$ symmetric positive definite matrix with $O(N^2)$ free parameters. Evaluating the Gaussian likelihood naïvely

requires $O(N^3)$ operations. Using a structured parametrization of the covariance matrix as the sum of a diagonal matrix and a low rank matrix, $\Sigma = D + VV^T$ where $D \in \mathbb{R}^{N \times N}$ is diagonal and $V \in \mathbb{R}^{N \times r}$, results in a compact representation with $O(N \times r)$ parameters. This allows the likelihood to be evaluated using $O(Nr^2 + r^3)$ operations. As the rank hyperparameter $r$ can typically be chosen to be much smaller than $N$, this leads to a significant speedup. In all our low-rank experiments we use $r = 10$. We investigate the sensitivity to this parameter of accuracy and speed in the Appendix.

Recall from Eq. 1 that $\mathbf{h}_{i,t}$ represents the state of an LSTM unrolled with values preceding $z_{i,t}$. In order to define the mapping $\Sigma(\mathbf{h}_t)$, we define mappings for its components

$$\Sigma(\mathbf{h}_t) = \begin{bmatrix} d_1(\mathbf{h}_{1,t}) & & 0 \\ & \ddots & \\ 0 & & d_N(\mathbf{h}_{N,t}) \end{bmatrix} + \begin{bmatrix} \mathbf{v}_1(\mathbf{h}_{1,t}) \\ \ldots \\ \mathbf{v}_N(\mathbf{h}_{N,t}) \end{bmatrix} \begin{bmatrix} \mathbf{v}_1(\mathbf{h}_{1,t}) \\ \ldots \\ \mathbf{v}_N(\mathbf{h}_{N,t}) \end{bmatrix}^T = D_t + V_t V_t^T.$$

Note that the component mappings $d_i$ and $\mathbf{v}_i$ depend only on the state $\mathbf{h}_{i,t}$ for the $i$-th component time series, but not on the states of the other time series. Instead of learning separate mappings $d_i, \mathbf{v}_i$, and $\mu_i$ for each time series, we parametrize them in terms of the shared functions $\tilde{d}, \tilde{\mathbf{v}}$, and $\tilde{\mu}$, respectively. These functions depend on an $E$-dimensional feature vector $\mathbf{e}_i \in \mathbb{R}^E$ for each individual time series. The vectors $\mathbf{e}_i$ can either be features of the time series that are known a priori, or can be embeddings that are learned with the rest of the model (or a combination of both).

Define the vector $\mathbf{y}_{i,t} = [\mathbf{h}_{i,t}; \mathbf{e}_i]^T \in \mathbb{R}^{p \times 1}$, which concatenates the state for time series $i$ at time $t$ with the features $\mathbf{e}_i$ of the $i$-th time series and use the following parametrization:

$$\mu_i(\mathbf{h}_{i,t}) = \tilde{\mu}(\mathbf{y}_{i,t}) = \mathbf{w}_\mu^T \mathbf{y}_{i,t}$$
$$d_i(\mathbf{h}_{i,t}) = \tilde{d}(\mathbf{y}_{i,t}) = s(\mathbf{w}_d^T \mathbf{y}_{i,t})$$
$$\mathbf{v}_i(\mathbf{h}_{i,t}) = \tilde{\mathbf{v}}(\mathbf{y}_{i,t}) = W_v \mathbf{y}_{i,t},$$

where $s(x) = \log(1 + e^x)$ maps to positive values, $\mathbf{w}_\mu \in \mathbb{R}^{p \times 1}, \mathbf{w}_d \in \mathbb{R}^{p \times 1}, W_v \in \mathbb{R}^{r \times p}$ are parameters.

All parameters $\theta_\mu = \{\mathbf{w}_\mu, \mathbf{w}_{\tilde{\mu}}\}$, $\theta_\Sigma = \{\mathbf{w}_d, W_v, \mathbf{w}_{\tilde{d}}\}$ as well as the LSTM update parameters $\theta_h$ are learned by optimizing Eq. 5. These parameters are shared for all time series and can therefore be used to parametrize a GP. We can view the distribution of $\mathbf{x}_t$ as a Gaussian process evaluated at points $\mathbf{y}_{i,t}$, i.e. $x_{i,t} = g_t(\mathbf{y}_{i,t})$, where $g_t \sim \text{GP}(\tilde{\mu}(\cdot), k(\cdot, \cdot))$, with $k(\mathbf{y}, \mathbf{y}') = \mathbb{1}_{\mathbf{y}=\mathbf{y}'} \tilde{d}(\mathbf{y}) + \tilde{\mathbf{v}}(\mathbf{y})^T \tilde{\mathbf{v}}(\mathbf{y}')$. Using this view it becomes apparent that we can train the model by evaluating the Gaussian terms in the loss only on random subsets of the time series in each iteration, i.e. we can train the model using batches of size $B \ll N$ as illustrated in Figure 2 (in our experiments we use $B = 20$). Further, if prior information about the covariance structure is available (e.g. in the case of spatial data the covariance might be directly related to the distance between points), this information can be easily incorporated directly into the kernel, either by exclusively using a pre-specified kernel or by combining it with the learned, time-varying kernel specified above.

## 5 Experiments

**Synthetic experiment.** We first perform an experiment on synthetic data demonstrating that our approach can recover complex time-varying low-rank covariance patterns from multi-dimensional observations. An artificial dataset is generated by drawing $T$ observations from a normal distribution with time-varying mean and covariance matrix, $\mathbf{z}_t \sim \mathcal{N}(\rho_t \mathbf{u}, \Sigma_t)$ where $\rho_t = \sin(t)$, $\Sigma_t = U S_t U^T$ and

$$S_t = \begin{bmatrix} \sigma_1^2 & \rho_t \sigma_1 \sigma_2 \\ \rho_t \sigma_1 \sigma_2 & \sigma_2^2 \end{bmatrix}$$

The coefficients of $\mathbf{u} \in \mathbb{R}^{N \times 1}$ and $U \in \mathbb{R}^{N \times r}$ are drawn uniformly in $[a, b]$ and $\sigma_1, \sigma_2$ are fixed constants. By construction, the rank of $\Sigma_t$ is 2. Both the mean and correlation coefficient of the two underlying latent variables oscillate through time as $\rho_t$ oscillates between -1 and 1. In our experiments, the constants are set to $\sigma_1 = \sigma_2 = 0.1, a = -0.5, b = 0.5$ and $T = 24,000$.

In Figure 3, we compare the one-step-ahead predicted covariance given by our model, i.e. the lower triangle of $\Sigma(\mathbf{h}_t)$, to the true covariance, showing that the model is able to recover the complex underlying pattern of the covariance matrix.

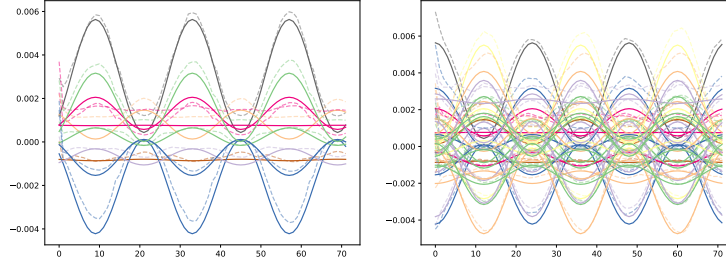

Figure 3: True (solid line) and predicted (dashed line) covariance after training with $N = 4$ (left) and $N = 8$ (right) time series. Each line corresponds to an entry in the lower triangle of $\Sigma_t$ (including the diagonal, i.e. 10 lines in the left plot, 28 in the right).

**Experiments with real-world datasets.** The following publicly-available datasets are used to compare the accuracy of different multivariate forecasting models.

- Exchange rate: daily exchange rate between 8 currencies as used in [16]
- Solar: hourly photo-voltaic production of 137 stations in Alabama State used in [16]
- Electricity: hourly time series of the electricity consumption of 370 customers [6]
- Traffic: hourly occupancy rate, between 0 and 1, of 963 San Francisco car lanes [6]
- Taxi: spatio-temporal traffic time series of New York taxi rides [31] taken at 1214 locations every 30 minutes in the months of January 2015 (training set) and January 2016 (test set)
- Wikipedia: daily page views of 2000 Wikipedia pages used in [11]

Each dataset is split into a training and test set by using all data prior to a fixed date for the training and by using rolling windows for the test set. We measure accuracy on forecasts starting on time points equally spaced after the last point seen for training. For hourly datasets, accuracy is measured on 7 rolling time windows, for all other datasets we use 5 time windows, except for taxi, where 57 windows are used in order to cover the full test set. The number of steps predicted $\tau$, domain, time-frequency, dimension $N$ and time-steps available for training $T$ is given in the appendix for all datasets.

**Evaluation against baseline and ablation study.** As we are looking into modeling correlated time series, only methods that are able to produce correlated samples are considered in our comparisons. The first baseline is VAR, a multivariate linear vector auto-regressive model using lag 1 and a lag corresponding to the periodicity of the data. The second is GARCH, a multivariate conditional heteroskedasticity model proposed by [34] with implementation from [12]. More details about these methods can be found in the supplement.

We also compare with different RNN architectures, distributions, and data transformation schemes to show the benefit of the low-rank Gaussian Copula Process that we propose. The most straightforward alternative to our approach is a single global LSTM that receives and predicts all target dimensions at once. We refer to this architecture as Vec-LSTM. We compare this architecture with the GP approach described in Section 4, where the LSTM is unrolled on each dimensions separately before reconstructing the joint distribution. For the output distribution in the Vec-LSTM architecture, we compare independent[4], low-rank and full-rank normal distributions. For the data transformation we compare the copula approach that we propose, the mean scaling operation proposed in [26], and no transformation.

| baseline | architecture | data transformation | distribution | CRPS ratio | CRPS-Sum ratio | num params ratio |
|---|---|---|---|---|---|---|
| VAR | - | - | - | 10.0 | 10.9 | 35.0 |
| GARCH | - | - | - | 7.8 | 6.3 | 6.2 |
| Vec-LSTM-ind | Vec-LSTM | None | Independent | 3.6 | 6.8 | 13.9 |
| Vec-LSTM-ind-scaling | Vec-LSTM | Mean-scaling | Independent | 1.4 | 1.4 | 13.9 |
| Vec-LSTM-fullrank | Vec-LSTM | None | Full-rank | 29.1 | 44.4 | 103.4 |
| Vec-LSTM-fullrank-scaling | Vec-LSTM | Mean-scaling | Full-rank | 22.5 | 37.6 | 103.4 |
| Vec-LSTM-lowrank-Copula | Vec-LSTM | Copula | Low-rank | 1.1 | 1.7 | 20.3 |
| GP | GP | None | Low-rank | 4.5 | 9.5 | 1.0 |
| GP-scaling | GP | Mean-scaling | Low-rank | 2.0 | 3.4 | 1.0 |
| GP-Copula | GP | Copula | Low-rank | 1.0 | 1.0 | 1.0 |

Table 1: Baselines summary and average ratio compared to `GP-Copula` for CRPS, CRPS-Sum and number of parameters on all datasets.

The description of the baselines as well as their average performance compared to `GP-Copula` are given in Table 1. For evaluation, we generate 400 samples from each model and evaluate multi-step accuracy using the continuous ranked probability score metric [20] that measures the accuracy of the predicted distribution (see supplement for details). We compute the CRPS metric on each time series individually (CRPS) as well on the sum of all time series (CRPS-Sum). Both metrics are averaged over the prediction horizon and over the evaluation time points. RNN models are trained only once with the dates preceding the first rolling time point and the same trained model is then used on all rolling evaluations.

Table 2 reports the CRPS-Sum accuracy of all methods (some entries are missing due to models requiring too much memory or having divergent losses). The individual time series CRPS as well as mean squared error are also reported in the supplement. Models that do not have data transformations are generally less accurate and more unstable. We believe this to be caused by the large scale variation between series also noted in [26, 27]. In particular, the copula transformation performs better than mean-scaling for GP, where `GP-Copula` significantly outperforms `GP-scaling`.

The `GP-Copula` model that we propose provides significant accuracy improvements on most datasets. In our comparison CRPS and CRPS-Sum are improved by on average 10% and 40% (respectively) compared to the second best models for those metrics `Vec-LSTM-lowrank-Copula` and `Vec-LSTM-ind-scaling`. One factor might be that the training is made more robust by adding randomness, as GP models need to predict different groups of series for each training example, making it harder to overfit. Note also that the number of parameters is drastically smaller compared to Vec-LSTM architectures. For the traffic dataset, the GP models have 44K parameters to estimate compared to 1.1M in a Vec-LSTM with a low-rank distribution and 38M parameters with a full-rank distribution. The complexity of the number of parameters are also given in Table 3.

We also qualitatively assess the covariance structure predicted by our model. In Fig. 1, we plot the predicted correlation matrix for several time steps after training on the Taxi dataset. We following the approach in [18] and reconstruct the covariance graph by truncating edges whose correlation coefficient is less than a threshold kept constant over time. Fig. 1 shows the spatio-temporal correlation graph obtained at different hours. The predicted correlation matrices show how the model reconstructs the evolving topology of spatial relationships in the city traffic. Covariance matrices predicted over time by our model can also be found in the appendix for other datasets.

Additional details concerning the processing of the datasets, hyper-parameter optimization, evaluations, and model are given in the supplement. The code to perform the evaluations of our methods and different baselines is available at https://github.com/mbohlkeschneider/gluon-ts/tree/mv_release.

# 6 Conclusion

We presented an approach to obtain probabilistic forecast of high-dimensional multivariate time series. By using a low-rank approximation, we can avoid the potentially very large number of parameters of a full covariate matrix and by using a low-rank Gaussian Copula process we can stably optimize directly parameters of an autoregressive model. We believe that such techniques allowing to estimate high-dimensional time varying covariance matrices may open the door to several applications in anomaly detection, imputation or graph analysis for time series data.

| | CRPS-Sum | | | | | |
|---|---|---|---|---|---|---|
| dataset estimator | exchange | solar | elec | traffic | taxi | wiki |
| VAR | 0.010+/-0.000 | 0.524+/-0.001 | 0.031+/-0.000 | 0.144+/-0.000 | 0.292+/-0.000 | 3.400+/-0.003 |
| GARCH | 0.020+/-0.000 | 0.869+/-0.000 | 0.278+/-0.000 | 0.368+/-0.000 | - | - |
| Vec-LSTM-ind | 0.009+/-0.000 | 0.470+/-0.039 | 0.731+/-0.007 | 0.110+/-0.020 | 0.429+/-0.000 | 0.801+/-0.029 |
| Vec-LSTM-ind-scaling | 0.008+/-0.001 | 0.391+/-0.017 | 0.025+/-0.001 | 0.087+/-0.041 | 0.506+/-0.005 | **0.133+/-0.002** |
| Vec-LSTM-fullrank | 0.646+/-0.114 | 0.956+/-0.000 | 0.999+/-0.000 | - | - | - |
| Vec-LSTM-fullrank-scaling | 0.394+/-0.174 | 0.920+/-0.035 | 0.747+/-0.020 | - | - | - |
| Vec-LSTM-lowrank-Copula | **0.007+/-0.000** | **0.319+/-0.011** | 0.064+/-0.008 | 0.103+/-0.006 | 0.326+/-0.007 | 0.241+/-0.033 |
| GP | 0.011+/-0.001 | 0.828+/-0.010 | 0.947+/-0.016 | 2.198+/-0.774 | 0.425+/-0.199 | 0.933+/-0.003 |
| GP-scaling | 0.009+/-0.000 | 0.368+/-0.012 | **0.022+/-0.000** | **0.079+/-0.000** | **0.183+/-0.395** | 1.483+/-1.034 |
| GP-Copula | **0.007+/-0.000** | **0.337+/-0.024** | **0.024+/-0.002** | **0.078+/-0.002** | **0.208+/-0.183** | **0.086+/-0.004** |

Table 2: CRPS-sum accuracy comparison (lower is better, best two methods are in bold). Mean and std are obtained by rerunning each method three times.

| | input | | output | |
|---|---|---|---|---|
| | | independent | low-rank | full-rank |
| Vec-LSTM | $O(Nk)$ | $O(Nk)$ | $O(Nrk)$ | $O(N^2k)$ |
| GP | $O(k)$ | $O(k)$ | $O(rk)$ | $O(N^2k)$ |

Table 3: Number of parameters for input and output projection of different models. We recall that $N$ and $k$ denotes the dimension and size of the LSTM state.

## Footnotes

* Work done while being at Amazon Research

[2]Note that the model complexity scales linearly with the number of Monte Carlo samples.

[3]This makes the underlying assumption that the marginal distributions are stationary, which is violated e.g. in case of a linear trend. Standard time series techniques such as de-trending or differencing can be used to pre-process the data such that this assumption is satisfied.

[4]Note that samples are still correlated with a diagonal noise due to the conditioning on the LSTM state.

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
