[Supplementary Material]

# High-Dimensional Multivariate Forecasting with Low-Rank Gaussian Copula Processes Supplementary material

**David Salinas**
Naverlabs
david.salinas.pro@gmail.com

**Michael Bohlke-Schneider**
Amazon Research
bohlkem@amazon.com

**Laurent Callot**
Amazon Research
lcallot@amazon.com

**Roberto Medico**
Ghent University
roberto.medico91@gmail.com

**Jan Gasthaus**
Amazon Research
gasthaus@amazon.com

## Contents

## A    Multivariate Likelihood

The probability density function of a multivariate normal distribution can be expressed as

$$\phi_{\boldsymbol{\mu},\Sigma}(\mathbf{x}) = \frac{1}{\sqrt{(2\pi)^d|L|}} \exp\left(-\frac{1}{2}||L^{-1}(\mathbf{x}-\boldsymbol{\mu})||^2\right)$$

where $\boldsymbol{\mu} \in \mathbb{R}^d$ and $L \in \mathbb{R}^{d\times d}$ is the Cholesky factor of the covariance matrix $\Sigma = LL^T$. This form is particularly amenable to computation using common neural network frameworks, as we only need to compute the determinant of a triangular matrix and solve a triangular system, which can both be done in $O(d^2)$.

This approach works in modest dimensions, but the quadratic cost in computational time and number of parameters becomes prohibitive when considering larger dimensions. This issue can be avoided by utilizing a low-rank matrix $\Sigma = D + VV^T$, where $D \in \mathbb{R}^{d\times d}$ and diagonal, and $V \in \mathbb{R}^{d\times r}$ where $r \ll d$ is a rank hyper-parameter. This parametrization of the covariance matrix also arises from the *factor analysis* model [18, 13, 3, 12], i.e. as the marginal distribution of $\mathbf{x}$ under the latent variable model

$$\mathbf{y} \sim \mathcal{N}(0, I), \quad x \sim \mathcal{N}(V\mathbf{x}, D)$$

When the covariance matrix is restricted in this way, it has $O(dr)$ parameters, and the Gaussian likelihood can be computed in $O(dr^2 + r^3)$ time.

In particular, the log-likelihood $\log \phi_{\boldsymbol{\mu},\Sigma}(x)$ can be evaluated by first computing an $r$-by-$r$ matrix $C = I_r + V^T D^{-1} V$ in $O(dr^2)$ time, followed by computing its Cholesky decomposition $C = L_C L_C^T$ in $O(r^3)$ time. Using the matrix determinant lemma in [7] we can write

$$\begin{aligned}
\log|\Sigma| &= \log|D + VV^T| \\
&= \log|C| + \log|D| \\
&= 2\log|L_C| + \log|D|
\end{aligned}$$

which can be computed in $O(d + r)$ as $L_C$ and $D$ are triangular. Given $L_C$, the Mahalanobis distance $\mathbf{x}^T \Sigma^{-1} \mathbf{x}$ can also be computed efficiently. By the Woodbury matrix identity we have $\Sigma^{-1} = D^{-1} - D^{-1}VC^{-1}V^T D^{-1}$. We can then write,

$$\begin{aligned}
\mathbf{x}^T\Sigma^{-1}\mathbf{x} &= \mathbf{x}^T(D^{-1} - D^{-1}VC^{-1}V^T D^{-1})\mathbf{x} \\
&= \mathbf{x}^T D^{-1}\mathbf{x} - \mathbf{x}^T(D^{-1}VC^{-1}V^T D^{-1})\mathbf{x} \\
&= \mathbf{x}^T D^{-1}\mathbf{x} - \mathbf{y}^T C^{-1}\mathbf{y}, \text{ with } \mathbf{y} = V^T D^{-1}\mathbf{x} \\
&= \mathbf{x}^T D^{-1}\mathbf{x} - ||L_C^{-1}\mathbf{y}||^2
\end{aligned}$$

The first term of the final equality can be computed in $O(d)$ and the second term can be computed with back-substitution in $O(r^2)$, so that the total time is $O(dr^2 + r^3)$ and the number of parameters is $O(dr)$.

The factor analysis latent variable model is closely related to PCA [21]: If we restrict the diagonal matrix $D$ to a multiple of the identity matrix, $D = \psi I$, we obtain a probabilistic version of PCA, from which the classical PCA can recovered in the limit $\psi \to 0$. Previous work [19] has applied PCA as a preprocessing step for uncovering latent structure. Here we propose an end-to-end approach that learns the structure of the covariance matrix jointly with the time series model.

## B    Empirical CDF

The naive empirical CDF estimator can exhibit large variance and the following truncated estimator from [10] is used instead:

$$\tilde{F}_i(v) = \begin{cases} \delta_m & \text{if } \hat{F}_i(v) < \delta_m \\ \hat{F}_i(v) & \text{if } \delta_m \leq \hat{F}_i(v) \leq 1 - \delta_m \\ 1 - \delta_m & \text{if } \hat{F}_i(v) > 1 - \delta_m \end{cases},$$

where choosing $\delta_m = \frac{1}{4m^{1/4}\sqrt{\pi \log m}}$ strikes the right bias-variance trade-off [10].

Further, we add jitter noise at training when computing the mapping $f_i$ to smooth the CDF for discrete data.

## C  Effect of rank on low-rank approximation

The effect of the low-rank approximation is analyzed in Figure 1 and Table 1 on the electricity dataset. As expected, the negative log-likelihood training loss decreases as the rank $r$ of $V$ increases in Figure 1. Table 1 shows the impact of the rank on the training/test loss. While the training loss decreases as the rank is increased, our model reaches its best performance on the test dataset with rank values 32/64. For higher ranks (128 and 256), the difference between training and test loss increases. This indicates that the high rank models may over-fit to the training data due to the flexibility of high-rank covariance matrix approximation.

Figure 1: Training loss vs training time when increasing rank on the electricity dataset.

| rank | test NLL | train NLL |
|------|----------|-----------|
| 1 | -291.4+/-8.2 | -288.9+/-8.2 |
| 2 | -306.2+/-6.7 | -304.8+/-5.7 |
| 4 | -319.3+/-4.9 | -312.1+/-3.5 |
| 8 | -333.6+/-7.7 | -330.2+/-6.3 |
| 16 | -334.8+/-4.9 | -337.5+/-4. |
| 32 | **-341.8+/-6.8** | -345.2+/-17.0 |
| 64 | -338.5+/-10.9 | -360.5+/-10.7 |
| 128 | -326.6+/-20.1 | -393.7+/-26.1 |
| 256 | -238.0+/-38.4 | **-423.1+/-20.7** |

Table 1: Error metrics when evaluating on the electricity dataset with increasing rank. We show the mean +/- 95% confidence interval over five runs.

## D  Baseline additional description

The GARCH [22] (Generalize Orthogonal - Generalize Autoregressive Conditional Heteroskedasticity) model is a composit model providing dynamics for the conditional mean and conditional covariance matrix of a multivariate system. The model for the conditional mean is, here, an autoregressive model of order one,

$$z_{i,t} = \alpha_i + \beta_i z_{i,t-1} + \epsilon_{i,t}.$$

Define $\epsilon_t = [\epsilon_{1,t}, ..., \epsilon_{N,t}]'$. To predict the conditional covariance matrix, the GARCH model maps $\epsilon_t$ to a set of $F = \min(N, T)$ independent factors $\mathbf{f}_t = [f_{1,t}, ..., f_{F,t}]'$, $\epsilon_t = A\mathbf{f}_t$, where A is a time independent, invertible matrix of dimension $[N \times F]$. The conditional variance $\sigma_{j,t}^2$, $j = [1, ..., F]$ of each of the factors is modeled using independent GARCH-type models, here a GARCH(1,0):

$$\sigma_{j,t}^2 = \omega_j + \gamma_j \sigma_{j,t-1}^2.$$

The linear mapping $A$ and the factors $\mathbf{f}$ are estimated using the ICA method of [2, 24]. Let $\mathbf{f}_t = H_t^{1/2}\mathbf{x}_t$, where $\mathbf{x}_t$ is a vector of independent random variables with conditional mean zero and conditional variance one. $H_t$ is the diagonal matrix of conditional variances of the factors with diagonal $[\sigma_{1,t}^2, ..., \sigma_{F,t}^2]$. The conditional covariance matrix $\mathbf{z}_t$ is then given by $\Sigma_t = A'H_t A$. We use the GARCH implementation of [5].

The VAR model is a multivariate linear vector auto-regression using lag 1 and a lag $l$ where $l$ corresponds to the periodicity of the data,

$$\mathbf{z}_t = \boldsymbol{\mu} + B_1 \mathbf{z}_{t-1} + B_l \mathbf{z}_{t-l} + \boldsymbol{\epsilon}_t.$$

$\boldsymbol{\mu}$ is a vector of intercepts of dimension $[N \times 1]$, and $B_1$ and $B_l$ are parameter matrices of dimension $[N \times N]$. Letting $z_i = [z_{i,l+1}, ..., z_{i,T}]'$, $\mathbf{x}_t = [\mathbf{z}_{t-1}', \mathbf{z}_{t-l}']'$, and $X = [\mathbf{x}_{l+1}, ..., \mathbf{x}_T]'$, each individual equation of the model can be written in stacked form as

$$z_i = \mu_i + X\boldsymbol{\theta}_i + \boldsymbol{\epsilon}_i, \tag{1}$$

where $\mu_i$ is a scalar intercept parameter and $\boldsymbol{\theta}_i$ is a parameter vector of length $2N$.

The parameters of equation 1 are estimated by the Lasso implemented in [4] using the procedure described in [8]. The estimated parameters are the minimizers of the loss function

$$L(\mu_i, \boldsymbol{\theta}_i) = \frac{1}{T-l} \left\| z_i - \mu_i - X\boldsymbol{\theta}_i \right\|_{\ell_2}^2 + 2\lambda_i \left\| \boldsymbol{\theta}_i \right\|_{\ell_1},$$

where $\|.\|_{\ell_2}$ is the $\ell_2$-norm and $\|.\|_{\ell_1}$ $\ell_1$-norm. $\lambda_i$ is a tuning parameter whose selection procedure is explained below.

## E   Hyperparameter optimization

Parameters are learned with SGD using ADAM optimizer with batch of 16 elements, l2 regularization with 1e-8 and gradient clipped to 10.0. For all methods, we apply 10000 gradient updates in total and decay the learning rate by 2 after 500 consecutive updates without improvement.

Table 2 lists the parameters that are tuned as well as the value of hyper-parameters that are not tuned and kept constant across all datasets.

| HYPERPARAMETER | VALUE OR RANGE SEARCHED |
|---|---|
| learning rate | [1e-4, 1e-4, 1e-2] |
| LSTM cells | [10, 20, 40] |
| LSTM layers | 2 |
| rank | 10 |
| num eval samples | 400 |
| conditioning length $m$ | 100 |
| sampling dimension $B$ | 20 |
| dropout | 0.01 |
| batch size | 16 |

Table 2: Hyper-parameters values fixed or range searched in hyper-parameter tuning.

To tune hyper-parameters of RNN methods we perform a grid-search of 12 parameters on Electricity and Exchange for `Vec-LSTM-ind-scaling`. The best hyperparameter for a method is set as the hyperparameter having the best average rank for CRPS. The best learning-rate/number-cells found for `Vec-LSTM-ind-scaling` is 1e-3 / 40, as LSTM and GP baselines has many variations, we use the same hyperparameter for all variants.

The Lasso estimator of the `VAR` model has a single Hyperparameter $\lambda_i$ for each equation. The best value of the parameter is selected within the sequence of values considered by the path-wise coordinate descent algorithm [4]. $\lambda_i$ is in the range $[\lambda_{i,min}, \lambda_{i,max}]$, where $\lambda_{i,max}$ is the smallest value of $\lambda_i$ such that all penalized parameters of the VAR are set to zero while $\lambda_{i,min} = \varepsilon \lambda_{i,max}$ where $\varepsilon = 0.0001$ if $N < T$ and $\varepsilon = 0.01$ otherwise[4]. The best value of $\lambda_i$ the value in the sequence that minimizes a Bayesian Information Criterion [8].

For the `GARCH` model, we performed a search among all combinations of mean and variance model specifications. The mean models considered were: AR(1), AR(Seasonal), VAR(1), VAR(Seasonal). The models for the variance components considered were: GARCH(1,0), GARCH(1,1), fGARCH(1,0) and fGARCH(1,1) [5]. We found that the only specification able to consistently converge in even the smallest of our datasets was using AR(1) as the mean model and GARCH(1,0) for the variance components.

# F  Dataset details

| dataset | $\tau$ (num steps predicted) | domain | frequency | dimension $N$ | time steps $T$ |
|---|---|---|---|---|---|
| Exchange rate | 30 | $\mathbb{R}^+$ | daily | 8 | 6071 |
| Solar | 24 | $\mathbb{R}^+$ | hourly | 137 | 7009 |
| Electricity | 24 | $\mathbb{R}^+$ | hourly | 370 | 5790 |
| Traffic | 24 | $\mathbb{R}^+$ | hourly | 963 | 10413 |
| Taxi | 24 | $\mathbb{N}$ | 30-min | 1214 | 1488 |
| Wikipedia | 30 | $\mathbb{N}$ | daily | 2000 | 792 |

Table 3: Summary of the datasets used to test the models. Number of steps forecasted, data domain $\mathcal{D}$, frequency of observations, dimension of series $N$, and number of time steps $T$.

Datasets (or their processing) will be made available after publication. Table 3 shows the properties of the used datasets. We only describe the processing for Taxi as all other datasets have been used in previous publications. The dataset obtained from [20] is preprocessed with the following steps similarly to [17]:

- Data cleaning: removal of outliers in terms of average speed ($> 45.31$ mph), trip duration ($> 720$ minutes), trip distance ($> 23$ miles) and trip fare ($> 86.6$);
- Data reduction: the dataset is reduced to the most active areas by retaining the area bounded by (40.70,-74.07) and (40.84,-73.95), expressed as (latitude, longitude) pairs;
- Data binning: first, the data is binned over time, using a frequency of 30 minutes; afterwards, the data is aggregated spatially, by binning latitude and longitude on a grid with spatial granularity of 0.001;
- For each subregion in the spatial grid and within each 30 minutes interval, the number of pickups and dropoffs are summed;
- Data filtering: the least active areas are filtered out from the data, by retaining only areas with at least $80\%$ non-zero observations. This results in a total of $1214$ time series.

We use January 2015 for the training set and January 2016 for the test set as in [17].

# G  Metrics

## G.1  Continuous Ranked Probability Score (CRPS)

The *continuous ranked probability score* (CRPS) [11, 6] measures the compatibility of a probability distribution $F$ (represented by its quantile function $F^{-1}$) with an observation $y$. The *pinball loss* (or *quantile loss*) at a quantile level $\alpha \in [0, 1]$ and with a predicted $\alpha$-th quantile $q$ is defined as

$$\Lambda_\alpha(q, y) = (\alpha - \mathcal{I}_{[y<q]})(y - q). \tag{2}$$

The CRPS has an intuitive definition as the pinball loss integrated over all quantile levels $\alpha \in [0, 1]$,

$$\text{CRPS}(F^{-1}, y) = \int_0^1 2\Lambda_\alpha(F^{-1}(\alpha), y)\, d\alpha. \tag{3}$$

An important property of the CRPS is that it is a *proper scoring rule* [6], implying that the CRPS is minimized when the predictive distribution is equal to the distribution from which the data is drawn.

In our setting, we are interested in evaluating the accuracy of the prediction compared to an observation $\mathbf{z}_t \in \mathbf{R}^N$. To do so, we generate predictions in the form of 400 samples which allows to estimate the quantile function $F^{-1}$ predicted by the model.

We then report average marginal CRPS over dimensions and over predicted steps in Table 4, e.g. we report

$$\text{E}_{i,t}[\text{CRPS}(F_i^{-1}, z_{i,t})]$$

where $F_i^{-1}$ is obtained by sorting the samples drawn when predicting $z_{i,t}$.

| | CRPS | | | | | |
|---|---|---|---|---|---|---|
| dataset<br>estimator | exchange | solar | elec | traffic | taxi | wiki |
| VAR | 0.015+/-0.000 | 0.595+/-0.000 | 0.060+/-0.000 | 0.222+/-0.000 | 0.410+/-0.000 | 4.101+/-0.002 |
| GARCH | 0.024+/-0.000 | 0.928+/-0.000 | 0.291+/-0.000 | 0.426+/-0.000 | - | - |
| Vec-LSTM-ind | 0.020+/-0.001 | 0.480+/-0.031 | 0.765+/-0.005 | 0.234+/-0.007 | 0.495+/-0.002 | 0.800+/-0.028 |
| Vec-LSTM-ind-scaling | 0.013+/-0.000 | 0.434+/-0.012 | 0.059+/-0.001 | 0.168+/-0.037 | 0.586+/-0.004 | 0.379+/-0.004 |
| Vec-LSTM-fullrank | 0.610+/-0.096 | 0.939+/-0.001 | 0.997+/-0.000 | - | - | - |
| Vec-LSTM-fullrank-scaling | 0.377+/-0.115 | 1.003+/-0.021 | 0.749+/-0.020 | - | - | - |
| Vec-LSTM-lowrank-Copula | **0.009+/-0.000** | **0.384+/-0.010** | 0.084+/-0.006 | 0.165+/-0.004 | 0.416+/-0.004 | **0.247+/-0.001** |
| GP | 0.029+/-0.000 | 0.834+/-0.002 | 0.900+/-0.023 | 1.255+/-0.562 | 0.475+/-0.177 | 0.870+/-0.011 |
| GP-scaling | 0.017+/-0.000 | 0.415+/-0.009 | **0.053+/-0.000** | **0.140+/-0.002** | **0.346+/-0.348** | 1.549+/-1.017 |
| GP-Copula | **0.008+/-0.000** | **0.371+/-0.022** | **0.056+/-0.002** | **0.133+/-0.001** | **0.360+/-0.201** | **0.236+/-0.000** |

Table 4: CRPS accuracy metrics (lower is better, best two methods are in bold). Mean and standard error are reported by running each method 3 times.

| estimator | MSE | MSE-sum | num_params |
|---|---|---|---|
| VAR | - | - | - |
| GARCH | - | - | - |
| Vec-LSTM-ind | 17.0 | 52.3 | 13.6 |
| Vec-LSTM-ind-scaling | 1.3 | 1.3 | 13.6 |
| Vec-LSTM-fullrank | 801.8 | 1545.6 | 103.4 |
| Vec-LSTM-fullrank-scaling | 985.3 | 1937.5 | 103.4 |
| Vec-LSTM-lowrank-Copula | 4.7 | 10.4 | 19.2 |
| GP | 122.3 | 1080.7 | 1.0 |
| GP-scaling | 1.1 | 3.0 | 1.0 |
| GP-Copula | 1.0 | 1.0 | 1.0 |

Table 5: Baselines summary and average ratio compared to GP-Copula for MSE, MSE-Sum, and number of parameters on all datasets.

To account for joint effect, we also report CRPS-Sum where accuracy is measured on the predicted distribution of the sum, e.g.

$$\mathrm{E}_t[\mathrm{CRPS}(F^{-1}, \sum_i z_{i,t})]$$

Where $F^{-1}$ is obtained by first summing samples across dimensions and then sorting to get quantiles.

Integrals are estimated with 10 equally-spaced quantiles.

### G.2 Mean Squared Error (MSE)

The MSE is defined as the mean squared error over all time series, i.e., $i = 1, \ldots N$, and over the whole prediction range, i.e., $t = T - t_0 + 1, \ldots, T$:

$$\mathrm{MSE} = \frac{1}{N(T - t_0)} \sum_{i,t} (z_{i,t} - \hat{z}_{i,t})^2 \tag{4}$$

where $z$ is the target and $\hat{z}$ the predicted distribution mean. Tables 5 - 7 show the MSE results for the marginal MSE and the MSE-sum. The definition of MSE-sum is analogous to CRPS-sum.

## H Comparison with forecasting methods with diagonal covariance

We evaluated our approach against DeepAR [14] and MQCNN [23], which we believe are a fair representation of the state-of-the-art in deep-learning-based forecasting. We also compared with DeepGLO [16] on two datasets provided by the authors. Table 8 lists the results of this comparison. Note that none of these competing approaches models correlations across time series in their forecasts (DeepGLO only provides point forecasts).

Figure 2: Correlation matrix predicted by our model at four different equally spaced time-steps with $step = 6$ for all datasets in this study. Exchange rate is nearly homoscedastic and most correlations are close to 1, because all currencies are relative to US dollar and therefore highly correlated. The remaining datasets are clearly heteroscedastic. For example, solar has low correlation at night and electricity/traffic/taxi follow day-night cycles. Wikipedia also shows heteroscedasticity across time.

| | MSE | | | | | |
|---|---|---|---|---|---|---|
| dataset<br>estimator | exchange | solar | elec | traffic | taxi | wiki |
| VAR | 4.4e-2+/-2.2e-5 | 7.0e3+/-2.5e1 | 1.2e7+/-5.4e3 | 5.1e-3+/-2.9e-6 | - | - |
| GARCH | 4.0e-2+/-5.3e-5 | 3.5e3+/-2.0e1 | 1.2e6+/-2.5e4 | 3.3e-3+/-1.8e-6 | - | - |
| Vec-LSTM-ind | 3.9e-4+/-2.0e-4 | 9.9e2+/-2.8e2 | 2.6e7+/-4.6e4 | 6.5e-4+/-1.1e-4 | 5.2e1+/-2.2e-1 | 5.2e7+/-3.8e5 |
| Vec-LSTM-ind-scaling | 1.6e-4+/-2.6e-5 | 9.3e2+/-1.9e2 | 2.1e5+/-1.2e4 | 6.3e-4+/-5.6e-5 | 7.3e1+/-1.1e0 | 7.2e7+/-2.1e6 |
| Vec-LSTM-fullrank | 5.2e-1+/-1.5e-1 | 3.8e3+/-1.8e1 | 2.7e7+/-2.3e2 | - | - | - |
| Vec-LSTM-fullrank-scaling | 6.5e-1+/-4.3e-2 | 3.8e3+/-6.9e1 | 3.2e7+/-1.1e7 | - | - | - |
| Vec-LSTM-lowrank-Copula | 1.9e-4+/-1.3e-6 | 2.9e3+/-1.1e2 | 5.5e6+/-1.2e6 | 1.5e-3+/-2.5e-6 | 5.1e1+/-3.2e-1 | 3.8e7+/-1.5e5 |
| GP | 3.0e-4+/-4.8e-5 | 3.7e3+/-5.7e1 | 2.7e7+/-2.0e3 | 5.1e-1+/-2.5e-1 | 5.9e1+/-2.0e1 | 5.4e7+/-2.3e4 |
| GP-scaling | 2.9e-4+/-3.5e-5 | 1.1e3+/-3.3e1 | 1.8e5+/-1.4e4 | 5.2e-4+/-4.4e-6 | 2.7e1+/-1.0e1 | 5.5e7+/-3.6e7 |
| GP-Copula | 1.7e-4+/-1.6e-5 | 9.8e2+/-5.2e1 | 2.4e5+/-5.5e4 | 6.9e-4+/-2.2e-5 | 3.1e1+/-1.4e0 | 4.0e7+/-1.6e9 |

Table 6: MSE accuracy metrics (lower is better). Mean and standard error are reported by running each method 3 times.

| | MSE-sum | | | | | |
|---|---|---|---|---|---|---|
| dataset<br>estimator | exchange | solar | elec | traffic | taxi | wiki |
| VAR | 1.2e0+/-8.6e-4 | 1.1e8+/-3.9e5 | 1.8e10+/-1.3e7 | 2.5e3+/-3.4e0 | - | - |
| GARCH | 1.1e0+/-2.0e-3 | 5.6e7+/-3.2e5 | 2.7e9+/-3.3e7 | 1.1e3+/-2.1e0 | - | - |
| Vec-LSTM-ind | 1.3e-2+/-7.0e-3 | 1.1e7+/-4.6e6 | 5.3e10+/-7.9e6 | 1.2e2+/-8.1e1 | 2.7e7+/-2.8e5 | 2.6e13+/-1.5e12 |
| Vec-LSTM-ind-scaling | 3.2e-3+/-1.3e-3 | 1.1e7+/-2.4e6 | 1.2e8+/-7.9e6 | 5.5e1+/-2.8e1 | 4.0e7+/-1.0e6 | 1.2e12+/-9.7e10 |
| Vec-LSTM-fullrank | 2.3e1+/-8.0e0 | 5.8e7+/-3.0e5 | 8.9e10+/-7.9e6 | - | - | - |
| Vec-LSTM-fullrank-scaling | 3.0e1+/-2.5e0 | 5.6e7+/-7.8e5 | 8.8e10+/-1.7e10 | - | - | - |
| Vec-LSTM-lowrank-Copula | 4.6e-3+/-8.3e-5 | 4.2e7+/-2.0e6 | 8.1e9+/-9.0e8 | 7.0e2+/-6.4e0 | 2.5e7+/-2.8e5 | 5.8e11+/-6.5e10 |
| GP | 7.2e-3+/-2.4e-3 | 5.5e7+/-1.0e6 | 8.6e10+/-1.9e9 | 4.3e5+/-2.2e5 | 3.3e7+/-1.8e7 | 3.5e13+/-9.5e10 |
| GP-scaling | 7.3e-3+/-1.8e-3 | 1.2e7+/-6.4e5 | 1.4e8+/-1.9e7 | 7.0e1+/-3.4e0 | 7.9e6+/-8.9e6 | 2.7e13+/-4.5e13 |
| GP-Copula | 4.2e-3+/-6.5e-4 | 1.2e7+/-8.3e5 | 1.5e8+/-3.5e7 | 6.2e1+/-4.3e0 | 1.0e7+/-1.1e6 | 1.9e12+/-2.2e15 |

Table 7: MSE-sum accuracy metrics (lower is better). Mean and standard error are reported by running each method 3 times.

| dataset | exchange | solar | elec | traffic | taxi | wiki |
|---|---|---|---|---|---|---|
| DeepAR [14] | **0.007** | 0.379 | 0.063 | 0.147 | **0.332** | 0.337 |
| MQCNN [23] | 0.013 | 0.482 | 0.078 | 0.177 | 0.657 | 0.277 |
| GP-Copula (Ours) | 0.008 | **0.371** | **0.056** | **0.133** | 0.360 | **0.236** |

| dataset | electricity | traffic |
|---|---|---|
| DeepGLO [16] | 0.109 | 0.221 |
| TRMF [16] | 0.105 | 0.210 |
| GP-Copula (Ours) | **0.083** | **0.168** |

Table 8: CRPS for additional baselines (left) and comparison with [16] when measuring WAPE (right).

# I  Predicted correlation matrices

We illustrate the learned correlations of our model on all datasets in Figure 2.

# J  Effect of the number of evaluation samples on CRPS and inference runtime

Figure 3 shows the effect of the number of evaluation samples on the CRPS and inference runtime. Drawing more than 100 samples only has a small effect on the CRPS and linearly increases the inference runtime.

# K  Additional experiments details

We use generic features to represent time. For hourly dataset, we use hour of day, day of week, day of month features. For daily dataset, we use day of week feature. For minutes granularity, we use minute of hour, hour of day and day of week features. All features are encoded with one number, for instance hour of day feature takes values in $[0, 23[$. Feature values are concatenated to the LSTM

(a) Effect of the number of evaluation samples (`num eval samples`) on the CRPS for electricity, solar, and taxi datasets. The line shows the mean CRPS over three independent runs and the shaded area shows the 95% confidence interval. Increasing the number of samples from 100 to 600 has a small effect on the CRPS (average CRPS over all datasets decreases from 0.272 to 0.271 for `GP-Copula` and from 0.52 to 0.50 for `Vec-LSTM-lowrank-Copula`, respectively).

(b) Effect of the number of evaluation samples (`num eval samples`) on the inference runtime. The inference runtime scales linearly with the number of drawn samples. The inferences runtimes for 10, 50, and 100 samples are similar due to initialization overhead. Note that the samples are only drawn during inference. Thus, the `num eval samples` parameter does not affect training runtime.

Figure 3: Effect of the number of evaluation samples on CRPS and inference runtime.

|  | CRPS-sum | | | | | |
|---|---|---|---|---|---|---|
| dataset<br>estimator | exchange | solar | elec | traffic | taxi | wiki |
| `GP-Copula` | **0.007+/-0.000** | **0.337+/-0.024** | **0.024+/-0.002** | 0.078+/-0.002 | 0.208+/-0.183 | 0.086+/-0.004 |
| `GP-Copula (GluonTS)` | **0.007+/-0.000** | 0.404+/-0.009 | 0.027+/-0.001 | **0.050+/-0.003** | **0.159+/-0.001** | **0.055+/-0.005** |

Table 9: CRPS-sum accuracy metrics for the GluonTS implementation of our model (lower is better). Mean and standard error are reported by running each method 3 times.

|  | CRPS | | | | | |
|---|---|---|---|---|---|---|
| dataset<br>estimator | exchange | solar | elec | traffic | taxi | wiki |
| `GP-Copula` | **0.008+/-0.000** | **0.371+/-0.022** | 0.056+/-0.002 | 0.133+/-0.001 | 0.360+/-0.201 | **0.236+/-0.000** |
| `GP-Copula (GluonTS)` | 0.009+/-0.000 | 0.416+/-0.007 | **0.054+/-0.000** | **0.106+/-0.002** | **0.339+/-0.001** | 0.244+/-0.003 |

Table 10: CRPS accuracy metrics for the GluonTS implementation of our model (lower is better). Mean and standard error are reported by running each method 3 times.

input at each time-step. We also lags values as input $\mathcal{L}$ according to the time-frequency, [1, 24, 168] for hourly data, [1, 7, 14] for daily, and [1, 2, 4, 12, 24, 48] for 30 minutes data.

All models are evaluated on a Amazon Web Services c5.4xlarge instance with 16 cores and 32GB RAM. All RNNs models take under five hours to perform training and evaluation. Missing numbers in Table 4 happens either because Out-of-memory prevents training or NaNs appear during training because of unstable models. Finally, RNNs are combined with Zone-out regularization [9] and residual connections and MXNet is used as the neural network framework [15].

## L    Open-source implementation of our model

We re-implemented the model described in this paper in GluonTS [1], an open-source time series toolkit. To ensure re-reproducibility, we released a static version of the code online that is not part of the latest GluonTS releases (for which we cannot guarantee reproducibility over time) at https://github.com/mbohlkeschneider/gluon-ts/tree/mv_release. Tables 9 and 10 show the benchmark results of our re-implementation. Our GluonTS implementation performs similar to the implementation that was used in this paper. In the new implementation, we set the `sampling dimension` $B$ to 2. Furthermore, we found that the piecewise-constant derivatives did not improve the results and removed them from our implementation.