[Reviews · NeurIPS 2019]

Reviewer 1



This paper designed a model with goal of forecasting on high dimensional time series data. To achieve its goal the model used LSTM network to capture the transition of latent states. In addition, to convert the latent states to observation domain, it has used the Gaussian copula process in which Gaussian process models a low rank covariance matrix which is computationally less complex to infer the parameters. Also the authors used Gaussian copula to convert the non-Gaussian observation to an standard Gaussian distribution. This will help them to enhance prediction power of the model since it converts non-Gaussian observation to have a standard Gaussian distribution. So we can summarize the contribution of this paper as following - The Paper tries to solve maximum likelihood problem with high dimensional observation domain. To use mini-batchs of data in training (use a few time series as mini-batch), authors propose Gaussian process models with low rank covariance matrix in which each observation have a non-time varying component which is learned by gaussian process and time varying component which is learned by the LSTM. This design makes the mini-batch learning possible. - To convert the non-gaussian and potentially heavy tail distributed data, authors used Gaussian copula to convert the non-Gaussian observation to a variable with standard Gaussian distribution. This will help them to enhance prediction power of the model. In experiments section, authors demonstrated application of their method using synthetic and real data and showed the proposed method have outperformed the competing auto regressive algorithms in many cases. Appendix contains details of experiment and hyperparameters setting for both propse model and competing algorithms. Quality Motivation, claims and and supporting material in main paper are explained well and the paper does not contain any significant theoretical or complicated design details . The quality of experimental results is good and all the hyper parameter setting and details of experiments have been explained well. Clarity: I think the paper objectives and explanation are pretty clear and flow of material is very smooth. There are some small issues needs to be fixed - line 132 R^d and R^d\times d → R^N and R^N\times N - supplementary line 6 remove the . From beginning of sentence Originality: As mentioned in summary the main contribution of this paper could be summarized as bellow - The Paper tries to solve maximum likelihood problem with high dimensional observation domain. To use mini-batchs of data in training (use a few time series as mini-batch), authors propose Gaussian process models with low rank covariance matrix in which each observation have a non-time varying component which is learned by gaussian process and time varying component which is learned by the LSTM. This design makes the mini-batch learning possible. - To convert the non-gaussian and potentially heavy tail distributed data, authors used Gaussian copula to convert the non-Gaussian observation to a variable with standard Gaussian distribution. This will help them to enhance prediction power of the model. The paper does not seem to have enough original contribution. Authors mostly have adopted existing techniques (see references below ) and algorithms and combined them together without showing much interpretation or theoretical results. Also method does not have proposed any smart regularization or parameters setting technique to avoid over-fitting and/or under-fitting. for Gaussian copula and variable transformation: - Aussenegg, Wolfgang, and Christian Cech. "A new copula approach for high-dimensional real world portfolios." University of Applied Sciences bfi Vienna, Austria. Working paper series 68.2012 (2012): 1-26. -Liu, Han, et al. "High-dimensional semiparametric Gaussian copula graphical models." The Annals of Statistics 40.4 (2012): 2293-2326. for low rank parameter estimation: similar to Liu, Haitao, et al. "When Gaussian process meets big data: A review of scalable GPs." arXiv preprint arXiv:1807.01065(2018) - and those refrenced by author in line 52 Significance:  The experiment section shows extensive experiment and relative success in comparison to other competing algorithms, but I would have some concern about syntactic data. The synthetic data are simple periodic data with the same period along all dimensions and I had expected that predicted line follow the synthetic data much more closely. Also as main goal of the paper is to perform the superior forecasting, it will be fair that results will be compared to paper below since these two papers try to solve the same problem and goal of both are superior forecasting power. Sen, Rajat, Hsiang-Fu Yu, and Inderjit Dhillon. "Think Globally, Act Locally: A Deep Neural Network Approach to High-Dimensional Time Series Forecasting." arXiv preprint arXiv:1905.03806 (2019) (I know it will be difficult as author of this paper has not shared the code yet) The other state-space model that can be considered is following Johnson, Matthew, et al. "Composing graphical models with neural networks for structured representations and fast inference." Advances in neural information processing systems. 2016.

Reviewer 2



This paper proposed a global and local combination method to forecast the multivariate time series. From a global perspective, the authors used the Gaussian Copula function to characterize dependency between multivariate time series. This results in low-rank estimation. From a local perspective, several local RNN-based TSF models are combined to estimate the sparse parameters. The motivations and working mechanism are lucid. However, there are some weaknesses. 1. Experimental contrast is not enough. In Table 1 and Table 2, the conventional models, such as VAR and GARCH are weaker here. It is more significant to validate that the model in this paper with the same settings can improve the performance of several state-of-the-art deep learning models. 2. The number of Monte Carlo sampling for sparse estimation is unspecified since random sampling does not seem to formalize the temporal evolution of time series. In other words, the temporal dependency and time-varying distribution of each individual time series also should be considered; otherwise, the number of samples may be a core parameter which influences the effectiveness of this model. Besides, in the case of independent random sampling, simply using RNNs and LSTMs to fit the underlying forecasting functions is not convincing. Furthermore, excessive sampling may increase the complexity of the entire model, which is not explicitly mentioned in this paper.

Reviewer 3



- Originality: I am not that familiar with the space of high-dimensional forecasting with modern deep learning methods. - Quality: This paper appears technically sound, the ideas are sensible and the experiments do a good job empirically testing the approach. Like all empirical investigations, more can be done --- in this particular case, there were are some tuning parameters that could affect performance that require (e.g. time series embedding vector size M and MSE/log like or time-varying residual analysis). - Clarity: This paper is very clearly written and the details of the model and training algorithm are thoroughly described. Some questions below address clarity. Other specific questions and comments: - line 76: This was unclear to me --- the pieces are of size epsilon^{-N}? Or there are that many pieces? - line 82: "order of magnitude larger than previously reported" needs a citation and to be made more precise. - line 85: "a principled, copula-based approach" is vague --- what principles are you referring to? - line 105: "...LSTM is unrolled for each time series separately, but parameters are tied across times series" --- what assumptions about the data does this particular model constraint encode? - line 124: Are all training chunks of size T + \tau? How well does look-ahead forecasting perform when the number of steps is greater than or less than \tau? Can this be made more robust by increasing or decreasing training chunk sizes? - line 135: How are these empirical CDF marginal distributions specified within the model? Do these distributions describe the observed marginal distribution of time series data? Or do they model the distribution of the residual given the model mean? - line 141: How does the discretization level m (here m=100) affect speed and model prediction accuracy? - line 164: How important is the embedding e_i? How does forecasting perform as a function of feature vector size E? - line 175: I don't fully follow the logic --- why does this Gaussian process view enable mini-batches? - line 186: How long does the time series need to be to fit a large LSTM that captures the dynamic covariance structure? In this synthetic example, how does the approach deteriorate (or hold up) as T shrinks? - line 225: regarding CRPS: it would be nice to give a short, intuitive explanation of CRPS and how it is different from other metrics, like log likelihood or MSE? Why not report MSE and log likelihood a well? - Table 2: How are the CRPS-sum error bars being computed? - Line 242: Besides the larger test error and a higher number of parameters do you see other signs that the Vec-LSTM is over-fitting (e.g. train vs test error)?

[Author Response · NeurIPS 2019]

We thank all reviewers for their valuable comments and suggestions. To address the point raised by the first two
reviewers regarding the comparison against recent DL methods, we evaluated our approach against DeepAR [1] and
MQCNN [3], which we believe are a fair representation of the state-of-the-art in deep-learning-based forecasting. We
also compared with DeepGLO [2] on two datasets provided by the authors.

| dataset | exchange | solar | elec | traffic | taxi | wiki | | dataset | electricity | traffic |
|---|---|---|---|---|---|---|---|---|---|---|
| DeepAR [1] | **0.007** | 0.379 | 0.063 | 0.147 | **0.332** | 0.337 | | DeepGLO [2] | 0.109 | 0.221 |
| MQCNN [3] | 0.013 | 0.482 | 0.078 | 0.177 | 0.657 | 0.277 | | TRMF [2] | 0.105 | 0.210 |
| GP-Copula (Ours) | 0.008 | **0.371** | **0.056** | **0.133** | 0.360 | **0.236** | | GP-Copula (Ours) | **0.083** | **0.168** |

Table 1: CRPS for additional baselines (left) and comparison with [2] when measuring WAPE (right).

We hope these additional experiments demonstrate that the proposed approach is not only competitive against classical
statistical techniques, but also against state-of-the-art approaches. Also note that none of these competing approaches
models correlations across time series in their forecasts (in fact DeepGLO only provides point forecasts). The code for
running the benchmark will be released after publication to help the community to evaluate forecasting methods.

**Reviewer #2** *"The paper does not seem to have enough original contribution. Authors mostly have adopted existing*
*techniques (see references below) and algorithms and combined them together ..."* Thank you for the relevant references
– we will add them to the paper. While we agree that the individual ingredients of our technique (Gaussian copula
models, GPs with low-rank covariance matrices, RNN models for time series forecasting) have been proposed and
studied before, we believe that the way they are combined in our approach is novel and non-trivial, and addresses what
we believe to be a highly-relevant, practical problem, namely robust high-dimensional time series forecasting.

*"The synthetic data are simple periodic data, expected that predicted line follow the synthetic much more closely."*
While the plot of Cov coefficient is a smooth cyclic periodic signal, the observed data is very noisy, making it hard to
regress the signal: we added a plot of the raw series to better illustrate that the signal is very weak compared to the
noise to show how this task is difficult.

*"As main goal of the paper is to perform the superior forecasting, it will be fair that results will be compared to paper*
*below"* As mentioned above, DeepGLO [2] only produces point forecasts (not distributions) and does not deal with
the high-dimensional covariance matrices that we tackle in this paper. Further, the referenced paper was published on
arXiv only a week before the submission deadline. We have since contacted the authors and obtained the first two data
sets used in their evaluation. The preliminary results above indicate that our method outperforms their approach in the
simpler point forecasting setting.

**Reviewer #3** *"In Table 1 ... more significant to validate that the model in this paper with the same settings can*
*improve the performance of several state-of-the-art deep learning models."* We added a comparison with [1, 3, 2] to
represent SOTA in deep-learning forecasting, see Tab. 1.

*"The number of Monte Carlo sampling ... excessive sampling may increase the complexity of the entire model."* Runtime
complexity of prediction increases linearly with the number of samples (training is not affected as there is no sampling
at that stage). We added an explanation in the manuscript and ran an experiment with different numbers of samples,
characterizing the impact of this parameter on the model's performance.

**Reviewer #4** *"How are the CRPS-sum error bars being computed?"* By rerunning each method with three different
seed and reporting mean/std (this detail was inadvertently omitted in the submitted manuscript).

*"More detailed experiments for certain aspects of the algorithm tuning different constant choices (e.g. rank, marginal*
*discretization level, embedding vector size)."* The rank hyperparameter is investigated in the appendix where we show
that (as one would expect) using a larger rank decreases training error but increases test error due to overfitting. The
sensitivity of the method to the other hyperparameters relative to properties of the data is an aspect that would be
interesting to investigate further, but the extensive experiments required come at a significant hardware cost.

*"CRPS is nice, but MSE, loglike, and visualizing temporal patterns in residuals ..."* We added MSE and will add loglike
when possible (as some models cannot compute it). Visualizing the pattern of residuals is done in the appendix; we will
reference this more clearly in the main text and add the analysis for all datasets.

[1] David Salinas et al. Deepar: Probabilistic forecasting with autoregressive recurrent networks. 2017.

[2] Rajat Sen et al. Think globally, act locally: A deep neural network approach to high-dimensional time series
forecasting. 2019.

[3] Ruofeng Wen et al. A multi-horizon quantile recurrent forecaster. 2017.


[Meta-Review · NeurIPS 2019]

This paper developed a method for probabilistic forecasting of high dimensional time series. The method parameterizes a structured covariance matrix with an RNN and incorporated a copula to account for dependence between series with heavy tailed marginals. Some of the reviewers had criticisms of the experiments. Specifically, comparing the method to other state of the art models instead of simpler time series models. One of these models a reviewer asked for was DeepGLO. Another reviewer took issue with the number of Monte Carlo samples used on the computation time. In all cases the authors addressed these issues with new experiments with results reported in the authors' response One reviewer also was concerned about the originality of the paper. However, the authors addressed this reasonably well in their response and looking at the paper the proposed approach solves an interesting problem and isn't a trivial combination of the constituent ideas. Overall, the I think that the authors have satisfactorily addressed the reviewers' main concerns and can be accepted.